



# Weight analysis of dam break risk consequences influencing factors

Zongkun Li[1], Wei Li[1], Wei Ge[1]

[1]School of Water Conservancy and Environment Engineering, Zhengzhou University 450001, China

*Correspondence to: Wei Li (gewei@zzu.edu.cn)*

**Abstract.** There are numerous influencing factors of the risk consequences of dam break. Scientific and reasonable index system and its weight distribution is one of the key element for comprehensive evaluation of the dam-break risk. Taking into consideration of 20 factors, including hazards, exposure and vulnerability factors, the evaluation index system of the consequences of the dam break risk is constructed. Using the Statistical Cloud Model (SCM) to improve the entropy method, we establish the weight calculation model of the influencing factors of the dam break risk consequences. The results shows

that the top 5 factors with the highest weight are risk population, flood intensity, alert time, risk understanding and distance from the dam. Compared to the traditional algebraic weight calculation methods, the result is basically consistent with the algebraic weight distribution, and increases the range by 2.03 times, supporting a more scientific basis for recognizing and evaluating the dam break risk consequences.

**Keywords:** Weight, dam break, risk consequences, entropy method, statistical cloud model

**1 Introduction**

Comprehensive evaluation of the risk consequences of dam break is the overall description of the severity of the consequences of the dam failure (Ling et al., 2009).The factors that can influence the risk consequences of dam break are usually composed of 3 factors, namely, hazards, exposure and vulnerability (Okada, 2004; Smith, 2013). The vulnerability factors can be further divided into 4 aspects: loss of life, economic loss, social impact and environmental impact. From the

point of view of system science, dam break flood disaster system is a dynamic system with high dimension, complexity and uncertainty. It accords with the development trend of risk assessment research "from low dimensional linearity to complex high-dimensional nonlinearity", "from single scale to multi-dimensional space-time scale", "from single scenario to combined scenario", "from certainty to uncertainty" (Zou et al., 2013).

The previous research on the index system of risk consequences and its weight is not sufficient. The uncertainty of the

impact of dam failure is explored, and suggestions for research index system are given (Lee and Noh, 2003; Wagenaar et al., 2016). The relationship between the hazard influencing factors and relationship between exposure and vulnerability factors is very complicated and the different types of flood including dam break flood can cause different degree of life loss (Jonkman et al., 2018; Wisner and Uitto, 2009). The indirect loss index for nature disasters is introduced and their weight is calculated by the traditional algebraic method (Daniell et al., 2018). The DAMBREAK computer programme is utilized to analysis the



downstream environmental impact and present 21 influence receptors, but the weight distribution of them is too average (Colomer and Gallardo, 2008). The Statistical Cloud Model (SCM) is used on the qualitative and quantitative transformation to analyze the regional water safety systems, but is not combine with the weight calculation (Ren et al., 2017). In the quantitative evaluation of risk consequences, we need to consider the combined effects of various factors, in which weight is

a key part of it. The function of weight is to coordinate and balance the difference between the indexes. It is a measure to unify each index without considering the dimension difference between the indexes. In order to evaluate the risk consequences more comprehensively and objectively, many influencing factors are needed. However, too many indicators, more than 9 for example, will bring such problems like the difficulty in expert scoring and consistency test, and too average weight distribution.

In the course of calculating the weights, different methods have their own emphasis. For example, entropy weight method as one of the important methods of weight calculation, does not adequately consider the subjective opinions of experts. The analytic hierarchy process (AHP) is faced with the difficulty of consistency checking when dealing with the conditions of multiple factors (>9)(Su et al., 2016). When previous studies used the data of Statistical Cloud Model (SCM) to calculate weights, they had neglected the entropy when applied the SCM to convert the subjective opinions, resulting in the

imperfection of information utilization (Mithas et al., 2011; Wan et al., 2015). These mentioned defects all lead to lack of scientificity in the calculation of weight. This manuscript introduces the SCM, which can reflect the fuzziness and randomness, to improve the entropy method for analyzing the weight of influencing factors of dam break risk consequences. The scientific influencing factors' weight will provide an important basis for further research on the dam-break risk comprehensive evaluation and for the establishment and improvement of dam risk management theory.

## 2 Methods

### 2.1 Risk index system

The establishment of evaluation index system is a systematic process. Scientific and reasonable evaluation index system is the guarantee for accurate risk assessment of dam failure, and the evaluation result is helpful for later research. Influencing factors of dam-break risk consequences are many and complicated in both quality and quantity, direct and indirect

contribution, natural and social ways (Zhou et al., 2014). We choose the representative indicators as much as possible to reduce the mutual influence and derivative of the indicators. For example, the risk population is the most direct factor of life loss, we only set it in the life bearing bodies, even though it is influencing the economic and social aspects, but in indirect and less crucial way (Dutta et al., 2003). In the selection of the economic impact factors, the selection of GDP(Gross Domestic Product) per capita can better reflect the economic situation of the dam area. Compared with the GDP of the area,

it is more accurate. Similarly, the comprehensive ability of water environment, soil environment and social carrying capacity is also selected. Whether the established index system is scientific and reasonable is directly related to whether it can objectively reflect the nature of the vulnerability itself. On the basis of aforementioned factors and characteristic of dam-





break flood system, we establish the risk influencing factor index system scientifically and reasonably as shown in Fig. 1.

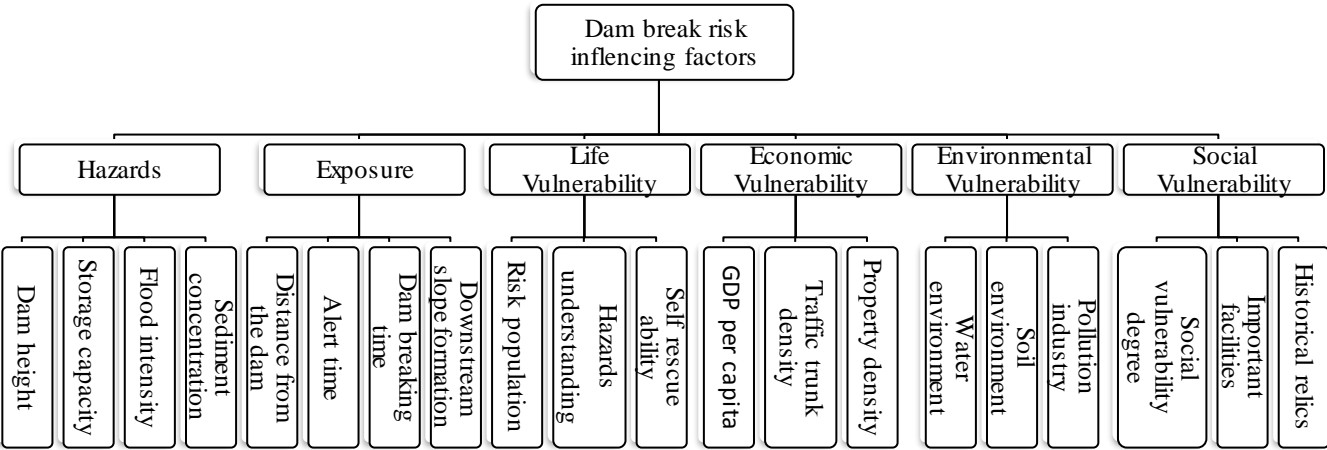

**Figure 1.** Index of dam-break risk consequences influencing factors.

## 2.2 Weight calculating model based on SCM-improved entropy method

Uncertainty is an intrinsic property of the objective world. The most important and most common uncertainties include fuzziness and randomness (Regas et al., 2010). The influencing factor system of the dam failure risk consequence is a multi-level and multi-index system with uncertainties (Li et al., 1995). In determining the importance of each risk factor to the comprehensive evaluation of the consequence, it needs a "quantitative conversion" of the uncertainty of the indicator. In the process of conversion, the expert's judgment makes a choice between many different factors that mutually affect each other and will absolutely lead to the ambiguity of boundaries, which is the fuzziness. On the other hand, the risk factors of dam break involve many aspects of life, economic loss, environmental and social impacts. In order to avoid the impact of expert's personal experience and subjective factors on the evaluation results, the risk factors of dam break need to adopt the method of group decision-making process. When an expert judged diverse risk factors, other experts must have some different opinions, reflecting in the randomness of judgments. Therefore, dam-break risk assessment system is a complex system integrating fuzziness and randomness. The SCM is invented under this context of the random and fuzzy feature of the dam break risk system. It describes the notions by the concept of clouds, reflects the randomness and fuzziness of concepts in natural language, and realizes the conversion between qualitative and quantitative information (Wang et al., 2016; Liu et al., 2018). In the process of group decision-making, the traditional method is only a simple algebraic operation of expert's ratings, which could not reflect the disagreements of different experts and the concentration of opinions. In fact, the experts' opinion is actually a rounded value that focuses on a certain degree of swing, which is using a stable tendency of the random number instead of the exact value, basically consistent with the central idea of SCM and the concept of entropy (Yari and Chaji,





2012; Wang et al., 2016).

### 2.2.1 SCM Theory

The SCM, which was proposed by Li Deyi, is a model of uncertainty transformation between a qualitative concept and quantitative numerical representation (Li et al., 1995; Li et al., 2014). It mainly reflects the fuzziness and randomness of the

5 concept of things or human knowledge in the objective world and integrates these two together. Constituting the mutual mapping between qualitative and quantitative, cloud generator is the key to its practical application.

Membership cloud: Suppose a universe $U=\{x\}$, $L$ is the language value of the link in $U$. The membership degree $R_L(x)$ of the element $x$ in $U$ to the qualitative concept expressed by $L$ is a stable random number. The membership degree distributed in the universe of discourse is called the membership cloud as shown in Fig. 2.

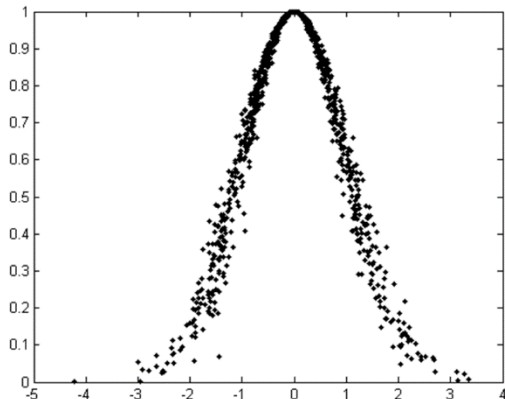

**Figure 2.** Sketch map of a Membership Cloud.

The $x$ and $y$ axes are for the expectation number and probability of distribution respectively. $R_L(x)$ takes a value between 0 and 1, whereas the cloud represents the mapping from the universe $U$ to the interval [0,1], that is: $R_L(x): U \rightarrow [0,1], \forall x \in U,$ $x \rightarrow R_L(x)$

15 It can be seen that the qualitative concept to the quantitative value on the universe $U$ is a one-to-many mapping relation, rather than a one-to-one relationship on the traditional fuzzy function. The degree of membership of $x$ to $L$ is a probability distribution, not a fixed value. SCM uses the expectation ($Ex$), entropy ($En$) and hyper entropy ($He$) as a whole to characterize an uncertain concept.

Expectation($Ex$): The mathematical expectation of cloud drop distribution in the universe of discourse, that is, the domain

20 value corresponding to the centric of the area under the coverage of the membership cloud, which is the domain value $x$ of the degree of membership. Generally, it is the point most capable of characterizing the qualitative concept, reflecting the information centre value of the corresponding fuzzy concept.

Entropy ($En$): $En$ is a measure of the ambiguity of a qualitative concept, reflecting the range of values that can be accepted



by the concept in the universe *U*. In the SCM, entropy is mainly used to measure the ambiguity and probability of qualitative concepts, reflecting the uncertainty of qualitative concepts. The larger the *En* is, the larger the range of values can be accepted by the concept and the more obscure the concept is. It embodies the flexibility of qualitative language.

Hyper-entropy (*He*): The measure of *En* uncertainty, entropy of entropy, reflects the discreteness of cloud drops. When the *He* is larger, the dispersion of cloud droplets is greater, that is, the greater the randomness of the membership value is, and the greater the "thickness" of the cloud can be. When it is closer to the concept centre or away from the centre, the randomness is relatively small, which is similar to a person's subjective feelings.

Cloud Generator: Generator is the most basic cloud algorithm, which can achieve quantitative range and distribution rules from the qualitative information expressed in language value. Cloud generators are mainly divided into the forward cloud generator and the backward cloud generator. The conversion process from qualitative concept to quantitative representation is conducted the forward cloud generator, the conversion process from quantitative representation to qualitative concept is produced by the backward cloud generator.

### 2.2.2 Entropy Method

The subjective weight analysis method is more dependent on the experts' opinions, and the consistency test under many factors is very difficult (Yari and Chaji, 2012). Therefore, this manuscript introduces entropy weight method as an objective weight calculation method. Entropy is a measure of uncertainty or randomness in information theory (Ouyang and Shi, 2013). In general, the more uncertain or random the event is, the more information it will contain, so the bigger entropy is. Therefore, the most important part of the entropy method is to obtain the differences in information, which is the degree of variation (Wang and Chen, 2016). According to the degree of variation of each index, we can calculate the entropy of each factor, and then use the entropy to adjust the weight of it, and finally, the objective weight value of the factors in the system is obtained.

The contribution of the numerical value in high-frequency or common consensus factor to the qualitative concept is greater than that of the numerical value in low-frequency (Yang and Nataliani, 2017). The En in the SCM could coincide with the idea of the entropy method in essence (Sheng et al., 2016; Dong et al., 2010). This manuscript makes use of the similar connotations of the SCM and entropy method. The objective advantages of entropy method need to be based on large amounts of score samples, which can be produced by the SCM cloud generator to get enough samples from limited experts' opinions. This manuscript attempts to use the SCM of qualitative-quantitative conversion model to improve the entropy method and make a scientific and objective response to the weight of risk factor.

### 2.2.3 Improved entropy method based on SCM

Suppose there are *n* indicators (column vectors) and *m* experts (row vectors). Each indicator computes the expectation and variance according to the cloud model. Statistical equation for calculating the *jth* indicator is as follow (Li et al., 1995):



$$Ex_j = \bar{x}_J = \frac{1}{m}\sum_{i=1}^{m} x_{ij} \qquad (i = 1, \dots, m; j = 1, \dots, n) \tag{1}$$

$$En_j = \sqrt{\frac{\pi}{2}}\frac{1}{m}\sum_{i=1}^{m}\left|x_{ij} - Ex_j\right| \qquad (i = 1, \dots, m; j = 1, \dots, n) \tag{2}$$

$$He_j = \sqrt{\frac{1}{m-1}\sum_{i=1}^{m}\left(x_{ij} - Ex_j\right)^2 - En_j^2} \qquad (i = 1, \dots, m; j = 1, \dots, n) \tag{3}$$

The weight equation for the indicator calculated with the use of the conventional algebraic method is as follow:

$$\omega_j = \frac{Ex_j}{\sum_{j=1}^{n} Ex_j} \qquad (j = 1, \dots, n) \tag{4}$$

This algebraic method is easy to use, but it does not make any use of the changes of $En$ in the SCM and may be misleading. For example, when the average scores of all indicators are the same, the weight of each indicator will calculate the same result. However, $En_j$ and $He_j$ could change greatly, but will not make enough reflection of the change in original equation, so an improved model is needed to replace this equation, as follow:

$$\widehat{\omega_j} = \begin{cases} \dfrac{Ex_j}{\ln(1+En_j)+1} \cdot \dfrac{1}{\sum_{j=1}^{n}\frac{Ex_j}{\ln(1+En_j)+1}} & ( En_j \neq 0 ) \\[4ex] \dfrac{Ex_j}{\sum_{j=1}^{n} Ex_j} & ( En_j = 0 ) \end{cases} \tag{5}$$

If the $En_j$ is not equal to 0, the equation of the weight is revised and the cloud entropy is involved in the calculation. The larger the cloud entropy, the more divergence of opinions the expert has on the index, so the weight of the index should be reduced. The smaller the entropy is, the smaller the expert's disagreement on the indicator, so the weight of the indicator should be increased. When the minimum entropy $En_j$ is equal to 0, indicating that the indicators of the experts have the same 15 score, and then the weight of the equation remained unchanged.

## 3 Results and discussion

### 3.1 Experts' scoring

According to the requirement of data volume based on entropy method, we invite 20 experts to score the index system. Each index scoring adopts 100 integral points system, according to the importance without any comparison between each other. 20 Scoring points should be scored from the perspective of comprehensive assessment of the risk consequences of the dam break. This scoring way can reveal the experts' opinion properly without imply any preference of the factors, and makes the scoring process easier. In accordance with the result of the score obtained by the backward cloud generator based on Eq. (1) to Eq. (3), its $Ex_j$, $En_j$, $He_j$ are obtained. In order to reflect the model characteristics of expert scoring more intuitively, the outstanding advantage of SCM, we present the sketch map of these 20 factors' membership cloud as in the Fig. 3:





Hazards (H)

H1:Dam height

(*Ex*:60.5 *En*:29.5 *He*:15.3 )

H2:Storage capacity

(*Ex*:79 *En*:16.8 *He*:3.6 )

H3:Flood intensity

(*Ex*:89 *En*:9.5 *He*:3 )

H4:Sediment concentration

(*Ex*:55.5 *En*:24.4 *He*:6.7 )

Exposure (E)

E1:Distance from the dam

(*Ex*:75.2 *En*:13.5 *He*:3.8 )

E2:Alert time

(*Ex*:80.2 *En*:10.6 *He*:1.2 )

E3:Dam breaking time

(*Ex*:60.5 *En*:10.5 *He*:2.3 )

E4:Downstream slope
formation

(*Ex*:59.3 *En*:18.4 *He*:7.6 )



| | | |
|---|---|---|
| Life vulnerability (Lv) | Lv1:Risk population (*Ex*:95.9 *En*:4.4 *He*:2 ) | Lv2:Hazards understanding (*Ex*:78.3 *En*:10.1 *He*:2 ) | Lv3:Self rescue ability (*Ex*:70.4 *En*:19.3 *He*:7.8 ) |

**Figure 3.** Sketch map of 20 indexes' membership cloud.

As shown in the Fig. 3, the centre vertex of the cloud is *Ex*, *En* represents the width of the cloud, and *He* represents the degree of dispersion of cloud distribution, that is, the thickness of cloud lines. For instance, the closer the *Ex* to the right side





of the axis, the higher the expert's score. The *En* of Lv3 is larger than that of Lv1, we can find the cloud is wider, and the *He* of E4 is larger than that of E2, so the distribution of the cloud is obviously "thicker" than E2. In a word, the membership cloud can obviously reflect the degree of divergence and randomness of expert opinions.

### 3.2 Weight calculation

5   After the result of the scoring is processed by the backward cloud generator according from the Eq. (1) to (3) and (5), the improved weight distribution result and result comparing the algebraic is shown in Table 1:

**Table 1.** Results comparison of the weight distribution.

| | Indexes | ω- (SCM-improved model) | ω- (Algebraic method) |
|---|---|---|---|
| | Risk population | 0.093 | 0.069 |
| | Flood intensity | 0.069 | 0.064 |
| | Alert time | 0.061 | 0.057 |
| | Hazards understanding | 0.060 | 0.056 |
| | Distance from the dam | 0.053 | 0.054 |
| | Storage capacity | 0.053 | 0.056 |
| | Social vulnerability degree | 0.051 | 0.050 |
| | Water environment | 0.048 | 0.047 |
| | Soil environment | 0.048 | 0.047 |
| **Weight** | Important facilities | 0.048 | 0.050 |
| | GDP per capita | 0.048 | 0.048 |
| | Traffic trunk density | 0.047 | 0.046 |
| | Pollution industry | 0.046 | 0.049 |
| | Dam breaking time | 0.046 | 0.043 |
| | Self rescue ability | 0.046 | 0.050 |
| | Downstream slope formation | 0.039 | 0.042 |
| | Historical relics | 0.037 | 0.044 |
| | Property density | 0.037 | 0.042 |
| | Dam height | 0.036 | 0.043 |
| | Sediment concentration | 0.034 | 0.040 |
| **Range** | | 0.059 | 0.029 |
| **Multiple** | | 2.044 | |

### 3.3 Discussion

In order to verify the validity of the method, the results of the distribution contrast of the original and improved ones are
10   drawn as Fig. 4.

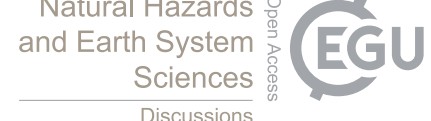

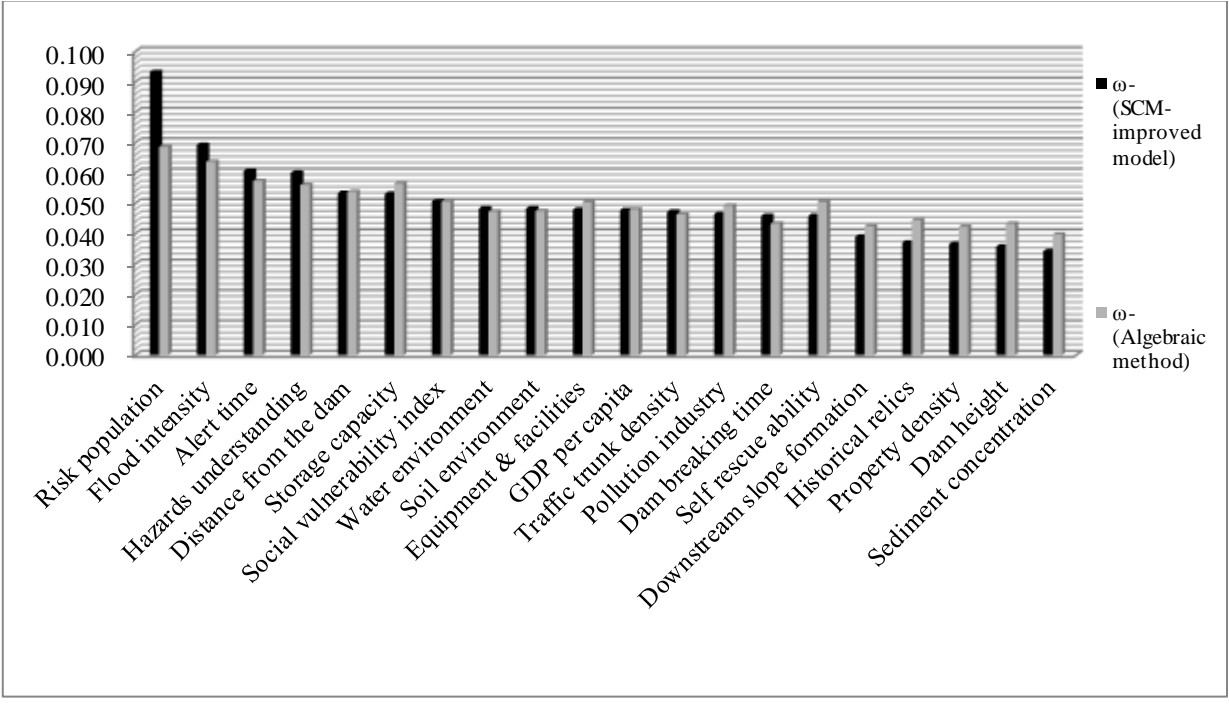

**Figure 4.** Weight comparing of 20 influencing factors.

According to the Fig. 3, Fig. 4 and Table 1, the analysis of figures shows following:

(1) The top rankings have not changed after the adjustment, and still maintain the consistency of ranking. All of these top ranking factors are scored higher and the opinions are concentrated, which is in line with the objective situation. At the same time, the range increased by 2.04 times.

(2) The distribution of weights is basically corresponding to the numerical value of *Ex*, reflecting the opinions of experts. At the same time, according to the adjustment of *En*, which reflects the difference of expert opinion, the weight of opinion unified index is further enlarged. Several factors are reduced the weight due to the large differences in opinions and the further reduction in adjusted weights. It reflects the validity of the entropy method in handling the weight distribution through the differences in opinions.

Thus, it can be seen that the SCM-improved entropy weight model is more in keeping with the general cognition of the people while ensuring the objective and fair data.

## 4 Conclusions

Dam break is a kind of low probability and high loss risk event with uncertainties. In this manuscript, risk factors are divided into hazards, exposure and vulnerability factors, and 20 factors are be selected as the main influencing factors of dam break risk consequences. We used SCM to improve the entropy method, based on the same ideas that these two methods are dealing with the divergence. The fuzziness index of the information is generated by the backward cloud generator, and then





applied to the improved formula of entropy weight calculation model. We establish the weight calculation model of influencing factors of dam break risk. The results indicate that (1) The result of weight calculation conforms to expert cognition, main factors' weight ranking is basically consistent with the one calculated by the traditional algebraic method. (2) Under the condition of 20 factors, the average problem of weight distribution is overcome, the difference between the

maximum and the minimum is 2 times larger. (3) This model is easy to apply and can be improved by the accumulation of expert opinions. In a word, it is reasonable and feasible to apply this improved model to the weight analysis of dam break risk factors, providing a solid foundation for risk assessment and risk management theory.

*Author contribution*. Zongkun Li and Wei Ge provided the funding acquisition and supervision. Wei Li developed the model
and wrote the original draft. Wei Li prepared the manuscript with contribution from all co-authors.

*Competing interests*. The authors declare that they have no conflict of interest.

*Acknowledgments*. The work was supported by National Natural Science Foundation of China (Grant No. 51379192,
51679222, 51709239) and Key Project of Science and Technology Research of Education Department of Henan Province of China (Grant No. 18A570007).

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
