# Peer review of "Weight analysis of influencing factors of dam break risk consequences"

_Natural Hazards and Earth System Sciences, 2018_

## Short Comment (SC1) · 19 Oct 2018

1. What makes the weight given by entropy weight method more reasonable? 2. In FIG. 4, the weight of the traditional method is higher and the weight of the new method is lower.

---

## Short Comment (SC2) · 30 Oct 2018

I wonder whether the experts were using the same order of magnitude to score each indicator. If yes, it is not easy to judge which of these indicators having the more impact on dam break under such big amount. Do you consider this problem and how to solve it?

---

## Short Comment (SC3) · 5 Nov 2018

The paper introduces an improved weight calculation model of the influencing factors of the dam break risk consequences combing with statistical cloud model and the entropy method, providing an innovative way for the dam risk management. The methodology proposed and well developed in this paper is judgmatic and improves the objectivity in evaluating the dam break risk consequences. In order to make the paper more suitable for the publication, I have following questions and suggestions which could be considered as the referential view points to improve the comprehensiveness.

1. Whether so many factors will cause the problem of decentralization of weight distribution?

[Figure]

2. Although the risk population is an important factor of loss of life, it will also cause considerable social impact. May the factor risk population be underestimated?

---

## Referee Comment (RC1) · Anonymous Referee #1 · 7 Nov 2018

This paper describes a study on the weight analysis of dam break risk consequences influencing factors. The topic is relevant to the journal and of interest to the readers. However, the paper has some issues to be resolved: 1) Please clarify the factors with more information. For example, what is flood intensity? What are the comprehensive abilities of water environment, soil environment and social carrying capacity? 2) Related to Point 1, how are these factors measured (or estimated)? 3) The study has classified the factors into hazards, exposure and vulnerability. Does it matter if a factor is wrongly classified into a category? Please explain its reason. 4) How could the results be validated? 5) There are some other studies on influential factors (such as "Calculation method and application of loss of life caused by dam break in China", Nature Hazards (2017) 85:39-57, Huang et al., DOI 10.1007/s11069-016-2557-9). Please

discuss your results with the published ones. 6) Please discuss why this study is useful to the stakeholders (e.g., local government, local communities), and how they could use the information.
* * *

---

## Author Comment (AC4) · 15 Nov 2018

Thank you for your good comments. Accoording to your comments, the response are as the following: 1. Admittedly, there is a dilemma in the calculation of factor weights: the weight of one or some factors may be too large to make other factors dispensable in calculating the result. However, if the distribution of factor weights is too average or scattered, the calculation of factor weights will lose its significance. This manuscript is different from the previous articles, either only for 3-4 first-level impact indicators, or only for a second-level indicators, considered 20 typical second-level indicators for the first time. From the results of this study, the new model does avoid the problem of too decentralized average weight, and the number of 20 does not cause the scattered problems. 2. On the issue of mutual interference between factors, experts

are required to mark the indicators based on the impact of indicators on the overall risk consequences. So, the social impact of population loss has been supposed to be taken into account, which would not cause the above mentioned problem. Welcome further discussion and thanks a lot.

---

## Editor Comment (EC1) · D. Patel (Editor) · 17 Nov 2018

Dear Reviewer(s), Thank you for reviewing the manuscript nhess-2018-265, entitled " Weight analysis of dam break risk consequences influencing factors " for NHESS.

We greatly appreciate the voluntary contribution that each reviewer gives to the Journal. We hope that we may continue to seek your assistance with the refereeing process for NHESS, and hope also to receive your own research papers that are appropriate to our aims and scope.

The decision that we arrived for this article was: Publish Subject to technical corrections, based the decision on your review and those of the other referees.

Yours Sincerely, Dr. Dhruvesh Patel Guest Editor, NHESS dhruvesh.patel@pdpu.ac.in

---

## Editor Comment (EC2) · D. Patel (Editor) · 17 Nov 2018

Dear Zongkun Li et al.,

Based on the advice received from reviewer(s), I have decided that your manuscript can be accepted for publication after you have carried out the corrections as suggested by the reviewer(s) and attached in revised manuscript, Therefore the decision that we arrived for this article was: Publish Subject to technical corrections.

Please make sure to submit your editable manuscript files (i. e. Word, PDF)."

Please submit your revised manuscript online by using the Editorial Manager system.

I am looking forward to receiving your revised manuscript in a one week period.

With kind regards,

Dr. Dhruvesh Patel Guest Editor, NHESS dhruvesh.patel@pdpu.ac.in

---

## Editor Comment (EC3) · Patel (Editor) · 26 Nov 2018

Discussions are so comprehensive and interesting; in result, its consequences are helpful to the researchers to understand the review process. I appreciate the effort of an eminent reviewer(s) for their brainstorming comments and also appreciate the author(s) to justify it rigorously and placed it delicately.

As per my opinion, paper will be accepted for publication after including all the technical suggestions by the reviewer in revise version, so at present it will be considered as major revision, considering technical revision.

I concern behalf of the stakeholder that the prescribed approach would be applicable for dam break cases before the break or after the break. Please include your comment

in revised version.

Is prescribed approach applicable only for china's dam? Or applicable for any dam break case in the world, kindly include your justification in conclusion part.

I am waiting to receive your revised manuscript in a week time.

—————————————

---

## Author Comment (AC5) · 29 Nov 2018

Dear Dr. Dhruvesh Patel:

Thank you for your precious comments. The responds to the reviewer's comments are as flowing:

Comments from Editor: 1) I concern behalf of the stakeholder that the prescribed approach would be applicable for dam break cases before the break or after the break. Please include your comment in revised version. Author's response: The applicability of the prescribed approach is very extensive. The main reasons are the flexibility of indicators selection and the independent feature of expert scoring. This manuscript analyzed the risk factors before the dam break. Different from risk analysis before the

dam-break, vulnerability indicators are the main focus for disaster loss analysis after the dam-break. So the stakeholders can select the vulnerability indicators only, and the approach of calculation is still effective. Because of the independence of expert scoring, the previous scoring records of experts are still available, which improves the efficiency of further weight evaluation.

2) Is prescribed approach applicable only for china's dam? Or applicable for any dam break case in the world, kindly include your justification in conclusion part. Author's response: The approach can be applicable to any countries or regions in the world, while the weight distribution in this manuscript is suitable for China. Because the weight calculation results depend on experts' scores, and experts from different countries have different understandings of risk factors, depending on their inclination. For example, in addition to the constant concern about the loss of life, some countries will regard environmental losses as more important than economic losses, while some countries will have different or even opposite opinions. So different countries can use this approach to get the weight distribution of risk factors suitable for their national conditions.

We already finished the revised version manuscript and the whole version of the Authors' Response. So we can upload them at the first time when the discussion is closed and the access is available. We want to thank you again for your efficient work.

Kind regards,

Wei Li Zhengzhou University on behalf of the co-authors

---

## Author Comment (AC6) · 3 Dec 2018

Dear Dr. Dhruvesh Patel,

We have already prepared the corresponding revisions and responses in accordance with your requirements. However, due to the system's automatic extended in the discussion process again, we cannot submit them on time. Please forgive us and we are looking forward for the completion of the review process. Thank you again for your efficient work.

Kind regards, Wei Li Zhengzhou University on behalf of the co-authors
* * *
[Figure]

2018-265, 2018.

---

## Author Response (AR1)

**Dear Dr. Dhruvesh Patel:**

Thanks very much for your kind work and consideration on publication of our paper. We have studied reviewer's comments carefully and have made revision which marked in the paper. We have tried our best to revise our manuscript according to the comments. The main corrections in the paper and the responds to the reviewer's comments are as flowing:

**1. Comments from Editor:**

 I concern behalf of the stakeholder that the prescribed approach would be applicable for dam break cases before the break or after the break. Please include your comment in revised version.

**Author's response:**

The applicability of the prescribed approach is very extensive. The main reasons are the flexibility of indicators selection and the independent feature of expert scoring. This manuscript analyzed the risk factors before the dam break. Different from risk analysis before the dam-break, vulnerability indicators are the main focus for disaster loss analysis after the dam-break. So the stakeholders can select the vulnerability indicators only, and the approach of calculation is still effective. Because of the independence of expert scoring, the previous scoring records of experts are still available, which improves the efficiency of further weight evaluation.

**Author's changes in manuscript:**

The extensive applicability advantage of the approach is added in the Conclusion section (P11, Line6-9).

2) Is prescribed approach applicable only for china's dam? Or applicable for any dam break case in the world, kindly include your justification in conclusion part.

**Author's response:**

The approach can be applicable to any countries or regions in the world, while the weight distribution in this manuscript is suitable for China. Because the weight calculation results depend on experts' scores and experts from different countries have different understandings of risk factors, depending on their inclination. For example, in addition to the constant concern about the loss of life, some countries will regard environmental losses as more important than economic losses, while some countries will have different or even opposite opinions. Therefore different countries can use this approach to get the weight distribution of risk factors which are suitable for their national conditions.

**Author's changes in manuscript:**

The borderless feature of the approach is added in the Conclusion section (P11, Line9-10).

**2. Comments from Anonymous Referee #1:**

This paper describes a study on the weight analysis of dam break risk consequences influencing factors. The topic is relevant to the journal and of interest to the readers.

- Please clarify the factors with more information. For example, what is flood intensity? What are the comprehensive abilities of water environment, soil environment and social carrying capacity?
- 2) Related to Point 1, how are these factors measured (or estimated)?

**Author's response:**

For question 1)and 2): Flood intensity reflects the approximate average damage caused by flood disasters, which can be jointly determined by the flood velocity and depth. The experience calculation formula often used is  $S_F = Q_{Top}/W_{Max}$ , thereinto,  $Q_{Top}$  is the peak discharge of dam break, and  $W_{Max}$  is the maximum width of the water surface formed by the flood.

Water environment and soil environment respectively refer to the quality of water and soil after being washed by dam-break flood. Their measurements can be based on the existing environment's vulnerability or sensitivity to flooding. For example, according to environmental functions and protection goals of surface waters, water environment can be divided into five categories in China in terms of functions, from source water to centralized domestic water to agricultural water. Soil quality can be divided into five categories, from desert (not suitable for vegetation growth) to woodland to national nature reserve.

The index of social comprehensive disaster bearing capacity includes the performance of downstream disaster response, disaster rescue and relief capacity, and post-disaster reconstruction capacity. Its calculation and evaluation require experts to comprehensively evaluate the indicators of different regions.

**Author's changes in manuscript:**

The definitions of the mentioned indicators are added in the Section 2.1 (P2, Line 30-34). The specific measurement of these indicators is not showed in manuscript because they are less relevant to the weight calculation and experts' scoring process.

**3**) The study has classified the factors into hazards, exposure and vulnerability. Does it matter if a factor is wrongly classified into a category? Please explain its reason

**Author's response:**

Hazards, exposure and vulnerability are only for the purpose of more orderly and logical elaboration of indicators, which will not affect the rating of the indicators by experts. Because the experts do not need to make a pair-wise comparison or consistency test on the indicators, they only need to judge and grade according to the importance of the indicators for the overall risk consequences. Taking this manuscript as an example, 20 indicators exist in parallel, and whichever category they belong to will have no any impact on the rating of experts.

**Author's changes in manuscript:**

The specific idea of expert scoring has been discussed and explained in Section 3.1 (P6, Line 24-26).

**4) How could the results be validated?**

**Authors' response:**

The verification of results can be verified from two aspects: one is the review based on expert experience; the other is the comparison with other methods. Taking this manuscript as an example, compared with the traditional method, the sorting is basically consistent, which further verifies the validity of the model. And the index weight is also roughly the same as Huang's article.

5) There are some other studies on influential factors (such as "Calculation method and application of loss of life caused by dam break in China", Nature Hazards (2017) 85:39-57, Huang et al., DOI 10.1007/s11069-016-2557-9). Please discuss your results with the published ones.

**Authors' response:**

As for Huang's article mentioned before, its main evaluation indicators are consistent with this result. The differences are that Huang's paper mainly studies the loss of life, while this manuscript studies the comprehensive consequences for the downstream; consequently the indicators are more abundant.

He did not compare the selected 11 indexes in the same dimension, but divides them to four different categories. If only in accordance with the sorting in a directory, sorting result is basically consistent, such as flood intensity > storage capacity, distance from the dam > dam breaking time, risk population > hazards understanding, and alert time > the downstream slope.

Furthermore, the indicators in this manuscript can be compared among different categories, not only within the specific category.

**Author's changes in manuscript:**

As the object of the study mentioned is focused on the loss of life, and the index system is only partly consistent with our manuscript, so it is not discussed in this manuscript.

6) Please discuss why this study is useful to the stakeholders (e.g., local government, local communities), and how they could use the information.

**Authors' response:**

The understanding of weight can help stakeholders to take more targeted measures to control risk factors, thereby reducing the overall risk. The index weight is an important basis for stakeholders to analyze the risk of their existing dam, because a considerable number of multi-index risk assessment models are functions or models related to the weight. In addition, the weight of indicators on the impact of risk consequences can also provide the stakeholders with the basis of fund allocation for reinforcement of risk prevention, so as to judge how to put funds and resources into the aspect with greater weight for risk management. In a word, the weight research can improve the effect of risk control and risk management.

**Author's changes in manuscript:**

The significance of factor weight analysis to stakeholders is added in the Section 4 (P11, Line 6-7).

**3. Comments from Public:**

1) What makes the weight given by entropy weight method more reasonable?

**Authors' response:**

Entropy weight method, as one of the objective weight analysis methods, has advantages over subjective evaluation method in avoiding the influence of experts' subjective opinions and dealing with multi-index problems. For example, the application of analytic hierarchy process (AHP) is limited by consistency testing, which usually deals with no more than 9 indicators. Ordinary relation analysis does not require consistency test, but it relies too much on the subjective judgment of experts, the relative importance of indicators will double the impact on the final results. Of course, entropy weight method also has its drawbacks, that is, there are more requirements for the evaluation of sample size, this paper is using the cloud model for simulation of expert scoring conversion, so as to meet the requirements of entropy weight method on the amount of data, and to improve the rationality and scientific of the calculation results.

2) In FIG. 4, the weight of the traditional method is higher and the weight of the new method is lower.

**Authors' response:**

According to table 4, the weight distribution and its trend of change are not entirely consistent. Some index weights increase (such as Lv1, H3, E2), some decrease (such as H1, H4, Ecv3), and some change little (such as E1, Sv1, Ecv1), which is the result of information entropy in the new model. Traditionally, when dealing with multi-index, one of the common problems is that the weight distribution is too average. Because of that, the new method is improved and the effect is obvious, and the range is increased by 104%.

**Author's changes in manuscript:**

We revised the content accordingly in this manuscript (Section 2.2.2 P5, Line 19-21).

3) I wonder whether the experts were using the same order of magnitude to score each indicator. If yes, it is not easy to judge which of these indicators having the more impact on dam break under such big amount. Do you consider this problem and how to solve it?

**Authors' response:**

Yes, the experts were using the same magnitude. There are 2 reasons for the fact that the factors in same magnitude and the full score system of 100 points is used. First, in order to ensure the objective and impartial of the experts, we want to get the most intuitive opinion from the expert without implying any preference. The second reason is to simplify the scoring difficulty of experts, because there are many factors and a wide range of fields involved. Unified dimension can simplify scoring process and avoid scoring errors due to difficulty. One of the advantages of this model is that it does not require experts to carry out consistency tests, reducing the difficulty of scoring. This model can deal with experts 'opinions into intuitive graphics, and with the continuous enrichment of experts, will further enhance the scientific nature.

**Author's changes in manuscript:**

We added some further explanation accordingly in section 3.1 (P6, Line 24-25).

4) Whether so many factors will cause the problem of decentralization of weight distribution?

**Authors' response:**

Admittedly, there is a dilemma in the calculation of factor weights: the weight of one or some factors may be too large to make other factors dispensable in calculating the result. However, if the distribution of factor weights is too average or scattered, the calculation of factor weights will lose its significance. This manuscript is different from the previous articles, either only for 3-4 first-level impact indicators, or only for a second-level indicators, considered 20 typical second-level indicators for the first time. From the results of this study, the new model does avoid the problem of too decentralized average weight, and the number of 20 does not cause the scattered problems.

**Author's changes in manuscript:**

The advantage of new model is added in section 3.3 (P10, Line 6).

5) Although the risk population is an important factor of loss of life, it will also cause considerable social impact. May the factor risk population be underestimated?

**Authors' response:**

On the issue of mutual interference between factors, experts are required to mark the indicators based on the impact of indicators on the overall risk consequences. So, the social impact of population loss has been supposed to be taken into account, which would not cause the above mentioned problem.

We have supplemented and revised the title, references and some other details of the manuscript, which do not affect the content and results of the article. The revisions are marked in yellow in the manuscript. On behalf of my co-authors, we would like to express our great appreciation to editor and reviewers.

Thank you and best regards,

Wei Li Corresponding author: Name: Wei Ge E-mail: gewei@zzu.edu.cn